# Effects of General Characteristics, Emotional Labor, Empathy Ability, and Wisdom on the Psychological Well-Being of Female Caregivers Visiting the Homes of Vulnerable Care Recipients and the Elderly

**DOI:** 10.3390/bs13050360

**Published:** 2023-04-26

**Authors:** Hee-Kyung Kim, Cheol-Hee Park

**Affiliations:** 1Department of Nursing, Kongju National University, Gongju 32588, Republic of Korea; 2Department of Nursing, Graduate School, Kongju National University, Gongju 32588, Republic of Korea

**Keywords:** emotional labor, empathy, wisdom, psychological well-being, caregivers, elderly

## Abstract

The purpose of this study is to analyze how general characteristics, emotional labor, empathy ability, and wisdom affect the psychological well-being of female caregivers. The research design is a descriptive correlational study. Data were collected using a self-report questionnaire, and analyzed using the SPSS Windows 27.0 program for hierarchical regression analysis. The results showed that there were differences in psychological well-being according to the work experience, education, and monthly income of 129 participants. In the analysis of the factors affecting the participants’ psychological well-being, model 1 showed 18.9% explanatory power with educational experience (ß = −0.23, *p* = 0.012) and monthly income (ß = 0.25, *p* = 0.007). In model 2, educational experience (ß = −0.23, *p* = 0.004), monthly income (ß = 0.20, *p* = 0.017), and emotional labor (ß = −0.41, *p* < 0.001) were the affecting factors, and the explanatory power increased by 16.1%, showing 35.0% overall. In model 3, educational experience (ß =−0.28, *p* < 0.001), emotional labor (ß = −0.35, *p* < 0.001), empathy ability (ß = 0.23, *p* = 0.001), and wisdom (ß = 0.52, *p* < 0.001) were the affecting factors, and the explanatory power increased by 36.9%, showing 71.9% overall. To enhance the psychological well-being of the participants, the head of the caregiving center should consider the education and income of caregivers. The center should also operate programs and prepare policies to reduce emotional labor and enhance empathy ability and wisdom.

## 1. Introduction

In 2022, the elderly population aged 65 or older accounted for 17.5% of the Korean population, and it is expected that this will continue to increase and reach 20.6% in 2025, making Korea a super-aged society [1]. Korea’s transition from an aging society to a super-aging society takes only 7 years, a faster transition compared to other major Organization for Economic Co-operation and Development (OECD) countries. There is a growing tendency among people to believe that the family, government, and society should all be responsible for supporting the elderly [1]. It is desirable for the government and society to support the vulnerable, such as the disabled and the elderly, in addition to their families.

Since 2008, the government has trained caregivers to provide professional care services for elderly individuals with difficulty performing daily tasks due to geriatric diseases such as dementia and stroke [2]. Caregivers are health professionals who offer physical and housekeeping support services in caring homes and facilities. They are part of an elderly home care center and work 24 h shifts at caring hospitals or centers to provide care services to the recipients in their homes. In Korea, the elderly have a culture of preferring to be cared for by caregivers while living in their homes, unless they have serious health conditions. Caregivers from the home care center often visit the homes of vulnerable care recipients and elderly individuals living in the community to provide proactive services such as with physical activity, housework, and daily life support. This helps recipients to maintain and improve their current physical functions and achieve emotional and psychological stability [3].

In particular, when it comes to visiting care, the caregiver is usually responsible for taking care of patients or elderly individuals who have poor physical or cognitive function. However, they may struggle with taking responsibility for the recipient’s health with accurate judgment and competent services. Therefore, it is necessary to help them maintain their psychological well-being, as this is the basis for providing a high-quality service. Female caregivers are required to have not only professionalism in areas such as care knowledge and elderly psychological knowledge, but also to perform physical and mental labor. As a result, they experience increased job stress, lower psychological well-being, and job satisfaction, which may cause them to even change jobs [4]. Especially during COVID-19, caregivers in charge of visiting care tend to experience decreased psychological well-being as stress levels increase [5]. Consequently, strategies are needed to improve their psychological well-being.

As a result of reviewing previous studies, this study aims to consider variables such as general characteristics, emotional labor, empathy ability, and wisdom as related factors to increase the psychological well-being of female caregivers responsible for visiting care.

Firstly, research should be conducted on which factors affect the psychological well-being of caregivers with regard to their demographic and job-related characteristics. The factors identified among general characteristics, including demographic characteristics and job-related characteristics, can be used as data to improve their psychological well-being in the future. Previous studies on female office workers have shown differences in psychological well-being depending on the level of education [5,6], economic status [7], income [6], and career [8]. Therefore, general characteristics, including these factors, have been analyzed to affect the psychological well-being of caregivers.

Second, emotional work experienced by caregivers in practice can harm their psychological well-being. A similar case can be found in professions such as bank clerks and nurses, which are traditionally considered women’s service professions. Morris and Feldman [9] defined emotional labor as the “effort, planning, and control needed to express organizationally desired emotion during interpersonal transactions”. In other words, the more emotional labor required, the lower the psychological well-being [10,11]. Accordingly, there is a need for interventions and institutional measures to reduce emotional labor among female caregivers [12]. In addition, caregivers experience emotional labor by directly or indirectly interacting with care recipients or their guardians. While providing direct care to individuals who need physical and mental support to maintain a normal daily life, caregivers perform the duties of emotional workers who provide labor and caring services with kind attitudes and compassionate actions under any circumstances [13].

Third, the ability to empathize can be cited as a significant factor that can improve psychological well-being. Empathy ability refers to a state of heightened awareness that arises when a person sympathizes with another person’s situation or condition [14]. It was defined as an emotional response that is more attuned to the other person’s situation than one’s own [15]. Empathy ability is crucial for understanding and providing help to others. For instance, the caregiver’s ability to empathize can mediate the relationship between the elderly’s communication and quality of service, thereby facilitating good communication and service [16]. Moreover, research suggests that the higher the empathy ability of college students, the better their psychological well-being [17], and that counselors’ empathy ability can affect their own psychological well-being [18]. Therefore, education aims to increase the empathy ability of female caregivers, thus improving their psychological well-being to bring about greater overall well-being.

In addition, since wisdom is a result obtained in the process of living life and is very important in making decisions and solving daily problems [19], it is necessary to include wisdom as one of these variables. Wisdom is the work of the mind to distinguish between good and evil, and it is considered the core of successful human development because it includes positive qualities such as self-integration and maturity, judgment and interpersonal skills, and an excellent understanding of life [20]. For office workers, the higher their wisdom, the higher their psychological well-being [21]. The wisdom of female caregivers was also found to be a significant factor in enhancing job competency [22]. In the relationship between emotional labor and the psychological well-being of bank tellers, wisdom was found to have a mediating effect [11]. Consequently, it can be inferred that if a female caregiver performs her duties wisely, she will gain psychological well-being.

Meanwhile, there are very few male caregivers currently working in the field, and their duties are different from those of female caregivers. Mainly, male caregivers work in long-term care hospitals, where they perform tasks such as bathing and moving male elderly patients, rather than providing home-visiting care. Overall, female caregivers were found to have higher stress, anxiety, and depression, and lower life satisfaction resulting in lower psychological well-being than male caregivers [23]. Accordingly, the purpose of this study was to analyze the factors that affect psychological well-being variables, including general characteristics, emotional labor, empathy, and wisdom. The hypothesis of this study was that the general characteristics, emotional labor, empathy ability, and wisdom of female caregivers will affect psychological well-being. The results of this analysis will be used as basic data for developing interventions to increase the psychological well-being of home-visiting caregivers.

## 2. Materials and Methods

### 2.1. Research Design

The research design is a descriptive correlational study that analyzes the effects of general characteristics, emotional labor, empathy, and wisdom on the psychological well-being of female caregivers.

### 2.2. Participants

The participants in this study were home-visiting female caregivers belonging to three elderly home-visit care centers located in D city, a large city.

The selection criteria were as follows: (1) adult females over 20 years old, (2) those who are currently providing home visit care, (3) those with more than 6 months of work experience, (4) those who signed the written consent form for this study and expressed their intention to participate in the study, and (5) those who can respond to the questionnaire and communicate effectively. The curriculum of caregivers encompasses a total of 240 h, with 80 h of theory related to caring, and 160 h of practical training. After obtaining the certificate, caregivers must have about six months of recognized practical experience to become true caregivers and facilitate job performance. To reduce confusion in the study’s results, the selection criteria were set as subjects with more than six months of practical experience [2].

To determine the number of participants required for this study, the author referred to a previous study on factors affecting turnover [11]. The analysis was conducted using the G-power3.1.9.6 program for statistical analysis of multiple regression analysis. The number of samples required to maintain 6 predictors, an effect size of 0.15, a significance level of 0.05, and a power of 0.90 was 123. Considering a dropout rate of 5%, the data for 129 participants were collected, and, finally, all 129 participants were selected to be included in the study.

### 2.3. Procedures

Data collection for this study was conducted over a period of four weeks, from January to February. Following approval from the Institutional Review Board of K University (IRB) (KNU_IRB_2022-143), the author and co-researcher visited three home-visit care centers for the elderly in D city. They explained the purpose and method of the study to the center director and collected data accordingly. The purpose of the study was to identify factors that affect psychological well-being using the general characteristics, emotional labor, empathy ability, and wisdom of female home-visiting caregivers. The researchers explained the study to the 129 female caregivers using a questionnaire. If a caregiver agreed to participate, she signed a written consent form and filled out a self-reported questionnaire that took approximately 15 min to complete. Permission for data collection was obtained from all participants. The questionnaire was designed to allow participants to ask any questions, and the researchers provided answers. Feedback on the results was provided to increase the study’s participation significance.

### 2.4. Measures

The instrument used for this study was a self-administered questionnaire consisting of 7 questions on general characteristics, 9 questions on emotional labor, 28 questions on empathy ability, 43 questions on wisdom, and 43 questions on psychological well-being.

#### 2.4.1. Emotional Labor

The emotional labor tool developed by Morris [9] was modified and supplemented by Song [24] and further modified by Kim [11] to suit the participants. This tool consists of a total of nine questions, which are divided into three subdomains: three questions on labor frequency, three questions on precautions for expressing emotions, and three questions on the dissonance of emotions. Each question is rated on a Likert 5-point scale, ranging from 1 point for ‘not at all’ to 5 points for ‘very much’, with an average score of 1–5; a higher score indicates higher levels of emotional labor. The reliability of Cronbach’s α in the study of Song [24] was 0.87, while in this study it was 0.86.

#### 2.4.2. Empathy Ability

The Interpersonal Reactivity Index (IRI) tool of Davis [25] was translated and corrected by Park [26]. The tool consists of a total of 28 questions, which are divided into 14 questions on cognitive empathy and 14 on emotional empathy. Each question is rated on a 5-point Likert scale ranging from 1 point for “not at all” to 5 points for “very much”.

Questions 3, 4, 7, 10, 12, 13, 14, 15, 17, 18, and 24 are inverse questions, which were inversely converted for the consistency of meaning. Higher scores mean higher empathy. The reliability of the tool during development was Cronbach’s α = 0.78; in this study, the reliability of cognitive empathy was 0.76, emotional empathy was 0.70, and overall reliability was 0.84.

#### 2.4.3. Wisdom

The Korean Wisdom Scale (KMWS) developed by Kim [27] was used. The tool consists of a total of 43 questions, which are divided into 4 subdomains: 16 questions for cognitive competence, 11 for refinement and balance, 10 for positive life attitude, and 6 for empathetic interpersonal relationships. Each question is rated on a 5-point Likert scale ranging from 1, “not at all” to 5, “very much”. The average score is 1–5 points, and the higher the score, the higher the level of wisdom. The reliability of the tool during development was Cronbach’s α = 0.93; in this study, the reliability of cognitive competence was Cronbach’s α = 0.97, refinement and balance was Cronbach’s α = 0.95, positive life attitude was Cronbach’s α = 0.95, empathetic interpersonal relationship was Cronbach’s α = 0.93, and overall reliability was Cronbach’s α = 0.98.

#### 2.4.4. Psychological Well-Being

The psychological Well-Being Scale (PWBS) developed by Ryff [28] and adapted by Kim et al. [29,30] into a Korean version was modified and used. The tool consists of a total of 43 questions, which are divided into six subdomains; seven questions on self-acceptance, seven on positive interpersonal relationships, seven on autonomy, seven on control over the environment, seven on life purpose, and eight on personal growth. Each question is on a Likert 5-point scale ranging from 1 point for ‘not at all’ to 5 points for ‘very much’, and higher scores indicating higher psychological well-being. Reliability of the tool during development was Cronbach’s α = 0.86 to 0.93, which in the study of Kim et al. [29], it was 0.66 to 0.76. In this study, the overall reliability was Cronbach’s α = 0.91, the subdomain of self-acceptance was Cronbach’s α = 0.82, the positive interpersonal relationship was Cronbach’s α = 0.83, the autonomy was Cronbach’s α = 0.59, the control over the environment was Cronbach’s α = 0.77, the purpose of life was Cronbach’s α = 0.71, and the personal growth was Cronbach’s α = 0.77.

### 2.5. Statistical Analysis

The SPSS Windows 27.0 program was used to analyze the degree of general characteristics and variables with descriptive statistics, such as real numbers, percentages, averages, and standard deviations. The comparisons between psychological well-being degree, according to the general characteristics, were analyzed with *t*-test, ANOVA, and Scheffe test. The correlation between participants’ emotional labor, empathy ability, wisdom, and psychological well-being was analyzed with Pearson’s correlational coefficients. The factors affecting the psychological well-being of participants were analyzed through hierarchical regression analysis using their general characteristics, emotional labor, empathy ability, and wisdom.

### 2.6. Ethical Considerations

This study was reviewed by the Institutional Review Board of K University regarding the research purpose, methodology, and human rights of the participants (KNU_IRB_2022-143). During the study period, guidelines on bioethics were followed. The consent form contained information about anonymity and confidentiality, and explained that the participant could voluntarily stop participating in the study at any time, even after agreeing to participate, and that there were no disadvantages to this. The form also explained the following details: the information collected will be managed in accordance with the Personal Information Protection Act, and authors will do their best to secure the confidentiality of all personal information obtained through the research; the collected data will be stored for 3 years in a locked cabinet accessible only to the authors, and will be computer coded and statistically analyzed while guaranteeing the anonymity of the participants; the data collected will be destroyed using the shredder after three years. A research statement was provided to the participants, and they provided their consent.

## 3. Results

### 3.1. General Characteristics of Participants

The study included 129 female caregivers who visited the care recipients’ homes. The participants’ age range was between 42 and 81 years, with an average age of 60 ± 7.76. There were 59 participants (45.7%) under the age of 59, and 59 (45.7%) between the ages of 60 and 69. Most of the participants were married (126, 97.7%), and 76 (58.9%) reported being non-religious. Regarding education, 100 (77.5%) participants had not graduated high school, and 73 (56.6%) had less than 1–5 years of work experience, with an average work experience of 4.84 ± 1.19 years. The majority of the participants (111, 86.0%) reported not having received any psychological well-being education within a year. In terms of income, 80 participants (62.0%) earned more than USD 1520 (KRW 2 million) per month (Table 1).

### 3.2. Differences in Psychological Well-Being According to Participants’ General Characteristics

Table 1 presents the differences in psychological well-being based on the participants’ general characteristics. The analysis found significant differences in psychological well-being among participants with respect to three variables: work experience (F = 7.64, *p* = 0.001), number of yearly psychological well-being education sessions (t = 6.58, *p* < 0.001), and monthly income (t = −4.17, *p* < 0.001).

Specifically, participants with more than 5 years of work experience had higher levels of psychological well-being than those with 1–5 years of work experience. Participants who did not receive psychological well-being education in the past year had higher levels of well-being than those who had received the education at least once. Finally, participants who earned more than USD 1520 per month had higher levels of psychological well-being than those who earned less than USD 1520 per month.

### 3.3. Emotional Labor, Empathy Ability, Wisdom, and Psychological Well-Being of Participants

Based on Table 2, the degree of the emotional labor for the participants was 3.05 ± 0.58 points out of 5 points, indicating a low-moderate level of emotional labor. The participants had a moderate level of empathy ability, with an overall score of 3.50 ± 0.35 out of 5 points. In terms of the subdomains, their cognitive empathy ability score was 3.52 ± 0.40 points, and their emotional empathy score was 3.48 ± 0.37 points. The participants also had an above-average degree of wisdom, with an overall score of 3.84 ± 0.58 points out of 5 points. The subdomain of cognitive competence score was 3.71 ± 0.69 points; temperance and balance score was 3.89 ± 0.59 points; positive life attitude score was 3.87 ± 0.60 points; and empathetic interpersonal relationships score was 4.02 ± 0.56 points. The degree of psychological well-being for the participants was moderate, with an overall score of 3.47 ± 0.30 points out of 5 points. The subdomain of self-acceptance score was 3.52 ± 0.42 points; positive interpersonal relationships score was 3.75 ± 0.63 points; autonomy score was low at 3.17 ± 0.38 points; control over the environment score was 3.70 ± 0.51 points; life purpose score was 3.71 ± 0.44 points; and personal growth score was 3.47 ± 0.54 points.

In other words, the participants’ emotional labor was low-moderate, empathy was moderate, and degree of wisdom was above average; however, the degree of wisdom was moderate-high in the mutual relationship. Psychological well-being was moderate, and psychological well-being in regard to autonomy was low.

### 3.4. Relationship between Participants’ Emotional Labor, Empathy, Wisdom, and Psychological Well-Being

The results of Table 3 suggest that there are significant relationships between emotional labor, empathy ability, wisdom, and psychological well-being among the participants. Specifically, the data show that there is a negative correlation between emotional labor (r = −0.45, *p* < 0.001) and psychological well-being, meaning that as emotional labor decreases, psychological well-being increases. Additionally, there is a positive correlation between empathy ability (r = 0.63, *p* < 0.001), indicating that higher levels of empathy ability are associated with greater psychological well-being. The results also indicate that cognitive empathy ability, emotional empathy ability, and total empathy ability are positively correlated with psychological well-being. This suggests that individuals who are more capable of understanding and experiencing the emotions of others have higher levels of psychological well-being.

Finally, the data revealed that there is a positive correlation between wisdom and psychological well-being. This suggests that individuals who possess greater wisdom tend to have higher levels of psychological well-being.

Overall, these findings suggest that emotional labor, empathy ability, and wisdom are important factors that influence psychological well-being. By understanding the relationships between these variables, it may be possible to develop interventions that promote greater well-being in individuals who are experiencing emotional difficulties.

### 3.5. Affecting Factors of Psychological Well-Being of Participants

Table 4 shows the results of analyzing the factors that affect the psychological well-being of the participants. Before conducting hierarchical regression analysis, this study used regression analysis to verify the basic hypothesis and examine the effects of emotional labor, empathy, and wisdom on psychological well-being. The Durbin–Watson test was performed to test for independence, and the result was 1.634, which is close to 2. Since there was no autocorrelation between the model error terms, the error terms satisfied the hypothesis of a normal distribution of residual differences. The P-P chart was also examined to verify the independence for the normality of the error term, which indicated that the error term showed a normal distribution. The tolerance limits were all within the range of 0.51–0.94, and the variance inflation factor (VIF) was within the range of 1.15–1.93, which did not exceed the reference value of 10, indicating that there was no problem with multicollinearity.

In the case of model 1, exogenous variables were controlled, and hierarchical regression analysis was performed by treating work experience, psychological well-being, education, and monthly income as dummy variables. As a result, education (β = −0.23, *p* = 0.012) and monthly income (β = 0.25, *p* = 0.007) had a significant effect, while the psychological well-being of female caregivers visiting the homes of care recipients was found to be significantly related to their work (F = 7.21, *p* < 0.001). The explanatory power of the above variables was 18.9%.

In the case of model 2, an additional 16.1% of the variance was explained by additionally inputting emotional labor as an emotional factor. Significant predictors included education (β = −0.23, *p* = 0.004), monthly income (β = 0.20, *p* = 0.017), emotional labor (β = −0.41, *p* < 0.001), and psychological well-being (F = 0.001). 13.23, *p* < 0.001). The explanatory power of these variables was 35.0%.

In the case of model 3, empathy ability and wisdom were added as social and cognitive factors, resulting in an additional 36.9% of the variance being explained, bringing the total to 71.9%. Significant predictors included education (β = −0.28, *p* < 0.001), emotional labor (β = −0.35, *p* <0.001), empathy ability (β = 0.23, *p* = 0.001), and wisdom (β = 0.52, *p* < 0.001), all of which had a significant effect on psychological well-being (F = 44.23, *p* < 0.001).

## 4. Discussion

This study attempted to analyze how general characteristics, emotional labor, empathy ability, and wisdom affect the psychological well-being of female caregivers. The participants’ main variable, psychological well-being, had a score of 3.47 ± 0.30 points out of 5 points in total. Positive interpersonal relationships, a subdomain, had the highest score of 3.75 ± 0.63 points, while autonomy had the lowest score of 3.17 ± 0.38 points. In a study of home-visiting caregivers during COVID-19, Kim et al. [29,30] found a psychological well-being level of 3.33 points, similar to the results of this study. As the psychological well-being score of caregivers is moderate and not high, the development and implementation of a psychological well-being promotion program for caregivers caring for the elderly with dementia and other vulnerable groups is necessary [30].

Among the participants’ general characteristics, differences in psychological well-being were observed based on work experience, number of psychological well-being education sessions, and monthly income. Although comparisons were difficult due to the lack of studies similar to this study, a previous study on female caregivers belonging to caring institutions (accounting for 96.3% of the participants) found that those with more than 5 years of work experience reported higher life satisfaction and happiness than those with less than 5 years of work experience, which supports the results of this study [31]. As work experience increases, self-concept for work increases, work performance improves, perspectives and degrees of adaptation to care recipients change, and confidence grows along with a sense of achievement based on experience, which can bring about psychological well-being [32]. Thus, it is necessary to improve the work environment of the centers, so that the participants can continue to be faithful to their duties without interrupting their careers. Moreover, the group without any experience of psychological well-being education exhibited a higher level of psychological well-being than those who received education more than once. Previous studies have shown that education for psychological well-being is effective. These include counseling programs [33] or positive psychology programs [34] that promote the happiness of full-time workers in education, which can reduce mental pain and improve happiness. However, the results of this study showed that the uneducated group had a high psychological well-being, so it is necessary to conduct a specific survey on education and repeat research on it to understand how to apply the program accordingly. Next, the level of psychological well-being was higher in the group earning a monthly income of USD 1520 or more compared to the group earning less than USD 1520. In a study conducted by Kim [6] on working women with children in the lower grades of elementary school, it was found that the group with a high monthly income had high psychological well-being, and that monthly income was a major factor affecting psychological well-being. In Nigeria, a positive correlation was found between caregiver income and psychological well-being; the higher the income, the higher the psychological well-being [35]. Among caregivers belonging to the visiting caring institutions, those with an income of USD 1460 or more reported higher happiness than those with less than USD 1460 [31], and female caregivers also acknowledged their higher economic status [7], which supports the results of this study. Consequently, it is important to conduct further research to review and improve caregivers’ salaries and incomes, so that they can receive appropriate compensation for their work, which in turn can improve their psychological well-being.

The study found a high correlation between the emotional labor, empathy ability, and wisdom of the participants and their psychological well-being. Emotional labor is a variable that draws continuous attention from service workers. Research has shown that the more emotional labor a female bank teller experiences, the lower her psychological well-being [11]. This study’s findings are consistent with that research. Female caregivers must conceal their feelings and show boundless kindness to patients as well as their guardians while caring for vulnerable care recipients or the elderly. However, if they fail to handle emotional labor properly, they may suffer from stress, as well as frustration, anger, and burnout. Strategies are thus needed to deal with negative emotions, while providing care. Empathy, a social factor, is positively correlated with psychological well-being. As a result of Kim’s study [16] on caregivers in charge of home-visit care, the higher the empathy ability, the higher the quality of service, and it was found that empathy had a mediating effect on the relationship between the communication and service quality of caregivers, which supported the results of this study. The ability to empathize with care recipients is ultimately judged to increase psychological well-being, and establishing positive relationships with others and performing care services well is associated with autonomy [28]; therefore, education to improve empathy for caregivers is required.

This study found that wisdom had a high correlation with psychological well-being, which was consistent with previous research. For example, a study on female service workers showed that wisdom can mediate the effects of emotional labor on psychological well-being [11], while another study on office workers identified wisdom as a major factor affecting psychological well-being [36]. Moreover, wisdom has been found to positively impact life satisfaction, physical health, family relationship quality, interpersonal relationships, and successful aging, due to its characteristics of self-preservation, judgement, and understanding [20,37]. In essence, wisdom enables individuals to integrate and comprehend diverse experiences, leading to better decision-making and adaptability, and consequently higher psychological well-being. Therefore, the study suggests that measures should be taken to promote caregivers’ wisdom development.

In terms of the factors affecting psychological well-being, the study conducted a hierarchical regression analysis and identified several main factors. Firstly, among general characteristics, the number of education sessions on psychological well-being in one year and monthly income were found to be significant factors. Secondly, when emotional labor was added as a psychological factor, education, monthly income, and emotional labor became the main factors, and explanatory power increased by 16.1%. Finally, when empathy ability and wisdom were included as social cognitive factors, education, emotional labor, empathy ability, and wisdom emerged as major factors, and explanatory power increased by 36.9% (Figure 1).

In the first step, the number of education sessions on psychological well-being in one year and monthly income were found to be the main factors affecting the psychological well-being of participants. Providing caregivers with conservative or continuous education on psychological well-being helps to increase psychological well-being. Conservative education can also bring ease to the job of caregivers [2]. Accordingly, educational contents should be customized and promoted in the form of development and application of intervention programs using surveys on psychological well-being, literature reviews, and various cases for nursing caregivers. Monthly income was consistent with other results reporting that it was a major influential factor on psychological well-being [6,8]. An economically stable life amid soaring prices positively affected the psychological well-being of nursing care workers. There is a difference in psychological well-being according to the monthly income of nursing care workers, and an appropriate compensation system can increase satisfaction with one’s job and bring happiness. It is thus necessary to improve the compensation system so that caregivers can be paid appropriately for their services [31].

In the second step, emotional labor was found to be the main factor affecting individuals. Emotional labor had a negative impact on interpersonal and communication skills, and it reduced autonomy and psychological well-being [38]. Therefore, female caregivers who work at the home elderly welfare center should manage emotional labor effectively because emotional labor tends to be severe, and their psychological well-being can decrease when they are visiting and caring for the elderly. In particular, female guardians have to make considerable emotional efforts because they must prioritize care recipients according to the organization’s policy while providing professional and qualitative care services. Additionally, they have a spirit of service and a sense of duty to overcome such emotional labor [39,40]. Consequently, it is essential to develop practical programs and establish systems to understand and resolve female caregivers’ emotional labor, control emotional inconsistencies, and increase job satisfaction and psychological well-being.

In the third step, empathy ability and wisdom were identified as important factors affecting the psychological well-being of the participants. Research by Oh [41] found that empathy ability was a factor affecting the psychological well-being of mothers with children in infancy. Empathy is beneficial in many situations, as people with high empathy tend to form high-trust relationships with others [42]. Specifically, since empathy can eliminate selfish views of oneself through the perspective of others, possessing empathy can help to reduce selfish and impulsive behaviors, eventually improving psychological well-being [43]. Therefore, increasing empathy ability among caregivers who assist vulnerable care recipients can lead to improved well-being for both the caregiver and recipients, as well as improved service quality and human relationships. In addition, wisdom was identified as the variable that had the greatest influence on female caregivers’ psychological well-being. A study by Webster [44] found that adult wisdom had a positive relationship with psychological well-being, and since wisdom is acquired through life experiences, it can increase with age [20]. Wise individuals can accept both positive and negative aspects of reality, and are aware of and accept the current reality [45], which can lead to improved life satisfaction and psychological well-being. Accordingly, it is important to find ways to increase wisdom among female vulnerable care recipients and the elderly.

Empathy ability showed a mediating effect in the relationship between the communication ability and quality of service provided by home-visiting caregivers, contributing to the improvement in service quality [16]. Wisdom showed a mediating effect in the relationship between emotional labor and psychological well-being of female service workers, which was an important factor in lowering emotional labor and increasing psychological well-being [11]. From this point of view, it is necessary to develop interventions that can increase psychological well-being by developing the empathy and wisdom of female caregivers.

The limitations and contributions of this study are as follows. Since the subjects selected for this study were a convenience sample, and their age range was limited to middle-aged and older adults, it is necessary to expand the region and participants, and include a wider range of ages. In addition, male caregivers have been excluded, so it is necessary to also study male caregivers. Using the results of this study, we identified education and income as the major factors, among the general characteristics, that contribute to the psychological well-being of female caregivers. This study also brought about theoretical expansion by including emotional labor, empathy ability, and wisdom. Furthermore, the results can be used for research on the psychological well-being of female caregivers and can be used as a basic data source for the development and application of intervention programs aimed at improving the psychological well-being of female caregivers by utilizing emotional labor, empathy ability, and wisdom.

## 5. Conclusions

The hypothesis of this study was that the general characteristics, emotional labor, empathy ability, and wisdom of female caregivers will affect their psychological well-being. As a result of identifying the factors affecting the psychological well-being of home-visiting caregivers, it was found that among the general characteristics, continuous education, monthly income, emotional labor, empathy ability, and wisdom were significant factors. Accordingly, in order to increase the psychological well-being of home-visiting female caregivers, it is necessary to develop and implement an intervention program to reduce emotional labor and improve empathy ability and wisdom. Additionally, if measures are taken to improve the continuous education and salary system of caregivers, this will help to increase the psychological well-being of female caregivers. This study provides fundamental data for establishing measures to improve the psychological well-being of home-visiting female caregivers in the future by identifying the factors involved. Moreover, it can be utilized as a fundamental data source by the heads of the centers in preparing systems and policies that can enhance the psychological well-being of female caregivers in the practice of caring for the elderly.

## Figures and Tables

**Figure 1 behavsci-13-00360-f001:**
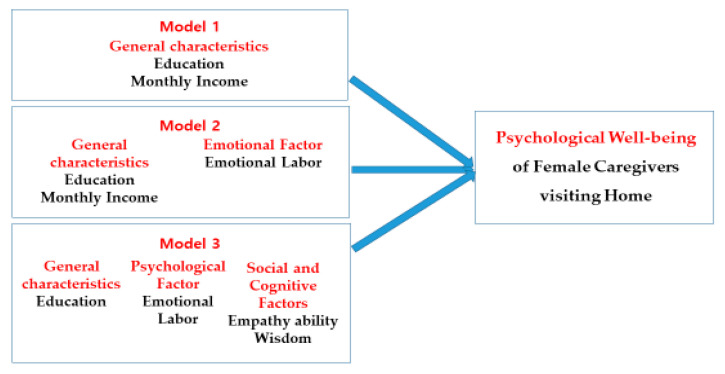
Factors affecting psychological well-being of female caregivers.

**Table 1 behavsci-13-00360-t001:** Differences in psychological well-being according to the general characteristics of female care workers (*N* = 129).

Variables	Classification	*n*	%	Psychological Well-Being Mean Interpretation SD			t/F	*p*-Value Scheffe Test
Age (years)	Below 59	59	45.7	3.57	moderate	0.38	1.49	0.230
	60–69	59	45.7	3.51	moderate	0.38		
	70 and over	11	8.56	3.36	low-moderate	0.38		
Marital status	Married	126	97.7	3.52	moderate	0.32	−0.17	0.867
	Unmarried, Divorced, etc.	3	2.3	3.57	moderate	0.72		
Religion	Religious	53	41.1	3.48	low-moderate	0.33	−1.07	0.287
	Non-Religious	76	58.9	3.55	moderate	0.40		
Level of education	Graduated from high school or lower	100	77.5	3.52	moderate	0.37	−0.16	0.874
	Graduated from college or higher	29	23.5	3.53	moderate	0.41		
Career (year)	Less than 1	8	6.2	3.40	low-moderate	0.28	7.64	0.001
	Between 1 and 5	73	56.6	3.63	moderate	0.39		b < c
	Over 5	48	37.2	3.38	low-moderate	0.31		
Number of education sessions on psychological well-being in one year	0	111	86.0	3.56	moderate	0.39	6.58	<0.001
(number)	Over 1	18	14.0	3.27	low-moderate	0.11		
Monthly income	Less than USD 1520 (KRW 2 million)	49	38.0	3.37	low-moderate	0.24	−4.17	<0.001
(USD)	USD 1520 (KRW 2 million) or more	80	62.0	3.61	moderate	0.41		

**Table 2 behavsci-13-00360-t002:** Degree of emotional labor, empathy, wisdom, and psychological well-being of the participants (*N* = 129).

Variables	Mean	SD	Interpretation	Actual Range
Emotional labor	3.05	0.58	low-moderate	1.67–5.00
Empathy ability	3.50	0.35	moderate	2.57–4.14
Cognitive empathy ability	3.52	0.40	moderate	2.36–4.14
Emotional empathy	3.48	0.37	moderate	2.57–4.43
Wisdom	3.84	0.58	moderate-high	2.63–4.95
Cognitive competence	3.71	0.69	moderate-high	2–5
Temperance and balance	3.89	0.59	moderate-high	2.45–5
Positive attitude to life	3.87	0.60	moderate-high	2.4–5
Empathetic interpersonal relationship	4.02	0.56	high	2.83–5
Psychological well-being	3.47	0.30	moderate	2.7–4.07
Self-acceptance	3.52	0.42	moderate	2.43–4.43
Positive interpersonal relationships	3.75	0.63	moderate-high	2.57–5
Autonomy	3.17	0.38	low-moderate	2–4.71
Control over the environment	3.70	0.51	moderate-high	2.57–5
Purpose of life	3.71	0.44	moderate-high	2.86–4.57
Personal growth	3.47	0.54	moderate	2.5–4.75

**Table 3 behavsci-13-00360-t003:** Relationship between participants’ emotional labor, empathy, wisdom, and psychological well-being.

Variables	Emotional Labor r (*p*)	Empathy Ability r (*p*)	Cognitive Empathy Ability r (*p*)	Emotional Empathy Ability r (*p*)	Wisdom r (*p*)	Psychological Well-Being r (*p*)
Emotional labor	1					
Empathy ability	−0.17 (0.062)	1				
Cognitive empathy ability	−0.20 (0.239)	0.92 (<0.001)	1			
Emotional empathy ability	−0.10 (0.239)	0.91 (<0.001)	0.69 (<0.001)	1		
Wisdom	−0.12 (0.164)	0.63 (<0.001)	0.50 (<0.001)	0.67 (<0.001)	1	
Psychological well-being	−0.45 (<0.001)	0.63 (<0.001)	0.52 (<0.001)	0.64 (<0.001)	0.66 (<0.001)	1

Level of significance: *p* < 0.05.

**Table 4 behavsci-13-00360-t004:** Factors affecting the psychological well-being of female care workers.

Variables	Model 1					Model 2					Model 3				
	B	SE	β	t	*p*	B	SE	β	t	*p*	B	SE	β	t	*p*
Constants	3.49	0.07		51.78	<0.001	4.31	0.16		26.73	<0.001	2.17	0.23		9.50	<0.001
Career (less than 1 year) (ref = less than 1 year)	−0.16	0.13	−0.10	−1.22	0.224	−0.22	0.12	−0.14	−1.82	0.071	−0.08	0.08	−0.05	−1.05	0.297
Career (over 5 years)	−0.11	0.08	−0.14	−1.44	0.153	−0.07	0.07	−0.09	−1.04	0.300	−0.05	0.05	−0.07	−1.12	0.263
Number of education sessions on psychological well-being in one year (1 time or more) (ref = none)	−0.24	0.10	−0.23	−2.55	0.012	−0.25	0.09	−0.23	−2.94	0.004	−0.30	0.06	−0.28	−5.24	<0.001
Monthly income (over KRW 2 million) (ref= less than KRW 2 million)	0.19	0.07	0.25	2.76	0.007	0.15	0.06	0.20	2.42	0.017	−0.06	0.05	−0.07	−1.28	0.205
Emotional labor						−0.27	0.05	−0.41	−5.52	<0.001	−0.22	0.03	−0.35	−6.96	<0.001
Empathy ability											0.24	0.07	0.23	3.44	0.001
Wisdom											0.34	0.04	0.52	7.74	<0.001
R^2^	0.189					0.350					0.719				
Adjusted R^2^	0.163					0.323					0.703				
Δ Adjusted R^2^ (*p*)						0.160 (<0.001)					0.380 (<0.001)				
F (*p*)	7.21 (<0.001)					13.23 (<0.001)					44.23 (<0.001)				

B = unstandardized coefficient Beta; SE = standardized error; β = standardization factor Beta; ref = reference.

## Data Availability

The data underlying this article will be shared upon reasonable request from the corresponding author.

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
