# Peer review of "Effects of General Characteristics, Emotional Labor, Empathy Ability, and Wisdom on the Psychological Well-Being of Female Caregivers Visiting the Homes of Vulnerable Care Recipients and the Elderly"

_behavsci, 2023, doi:10.3390/bs13050360_

Round 1
Reviewer 1 Report
Dear Authors,
Your study: Effects of Emotional Labor, Empathy Ability, and Wisdom on Psychological Well-being of Female Caregivers visiting Home of Vulnerable Care Recipients and the Elderly, is very interesting; however, I have noted the following in order to improve your paper:
1. The variables of the study which include emotional labor, empathy ability, wisdom, and psychological well-being should be clearly defined and discussed in the introduction, results, and discussion. It is difficult for the reader to follow the focus of the study because these variables have not been well-established in the introduction.
2. In the introduction, you discussed some variables on socio-demographic data but these were not mentioned in your title and in the aim/purpose of your study.
3. The last paragraph of the study mentioned about the study helping the vulnerable and the elderly recover their health. Is this part of your study or do you mean it as a result of improving the psychological well-being of the care givers?
4. Lines 56-57: "Moreover, most caregivers in charge of visiting care are women, who are more likely to experience emotional difficulties than men." kindly support this seemingly biased statement.
5. lines 66-67- are the caregivers nurses? If yes, then decide which term to use: caregivers or nurses? There is a difference between the terms caregivers and nurses. Kindly clarify.
6. I suggest that you re write your introduction by building a strong background on the gap regarding the psychological well-being of the caregivers followed by descriptions and discussions of the factors you have identified such as emotional labor, empathy ability, wisdom, and the socio-demographic profiles. It is recommended that you cite theoretical background/foundation of these concepts.
7. Procedures: Lines 136-139 - Kindly explain further what do you mean by the statement. It is not clear.
8. Instruments: Were the questions in English or translated to the vernacular? If yes, kindly describe the procedure of translation and how was it validated and tested for internal consistency?
9. Regression analysis was not mentioned under statistical analysis.
10. In table mean- the labels did not identify which is the psychological well-being. Is it the mean scores? If yes, what are the interpretation of the mean scores? Do they have low/poor, moderate, or high/good psychological well-being?
11. Table 1- I suggest that you include a dollar equivalent of the won currency. What do you mean by yes or no under religion? Also what do you mean by education received?
12. Lines 228-229 - Kindly review the statement.
13. Table 2- I suggest the you organize the scales by categorizing them according to your main variables to have a clear picture. Also to add an interpretation ( e.g. low, moderate, high) of the scores for each of the categories (emotional labor, empathy, wisdom, psychological well-being).
14. Table 3- what are the relationships?
15. Discussion- lines 295-308 - I do not find it relevant to the focus of your study. Your discussion should revolve around support on your hypothesis (correlations and predictive models to psychological well-being) and not deviate from the variables you have identified.
16. Figure 1- The model does not reflect the hypothesis and finding of the study.
17. The conclusion sounds like a summary of the results. Therefore, what can you conclude from these findings?
General comments:
The organization and grammar of the article should be considered. There are repetitive concepts, and grammar lapses that make many of the sentences confusing.
There should be consistency and clear link among the variables being studied, strong support of the study hypothesis, and consistency in following through from the introduction to the discussion.
Thank you and good luck.
Author Response
|
Reviewer 1 |
|
|
Your study: Effects of Emotional Labor, Empathy Ability, and Wisdom on Psychological Well-being of Female Caregivers visiting Home of Vulnerable Care Recipients and the Elderly, is very interesting; however, I have noted the following in order to improve your paper: 1. The variables of the study which include emotional labor, empathy ability, wisdom, and psychological well-being should be clearly defined and discussed in the introduction, results, and discussion. It is difficult for the reader to follow the focus of the study because these variables have not been well-established in the introduction. |
Thank you very much for your careful review to make it a good paper. As you pointed out, I revised it hard.
In the introduction, results, and discussion, the variables of the study including psychological well-being, emotional labor, empathy, and wisdom were clearly defined and the discussion was organized. Thank you for pointing that out. |
|
2. In the introduction, you discussed some variables on socio-demographic data but these were not mentioned in your title and in the aim/purpose of your study. |
Added at title and purpose Title: Effects of General Characteristics , Emotional Labor, Empathy Ability, and Wisdom on Psychological Well-being of Female Caregivers visiting Home of Vulnerable Care Recipients and the Elderly Purpose: Abstract: The purpose of this study is to analyze general characteristics, emotional labor, empathy ability, and wisdom affect psychological well-being for female care workers. |
|
3. The last paragraph of the study mentioned about the study helping the vulnerable and the elderly recover their health. Is this part of your study or do you mean it as a result of improving the psychological well-being of the care givers? |
Changed Therefore, the author first analyzes the general characteristics of home-visiting female caregivers, emotional labor as an emotional factor, and empathy ability and wisdom as social and cognitive factors to identify factors that affect psychological well-being. Through this, the results of this study aim to provide basic data for medium-term development to increase the psychological well-being of female caregivers. |
|
4. Lines 56-57: "Moreover, most caregivers in charge of visiting care are women, who are more likely to experience emotional difficulties than men." kindly support this seemingly biased statement. |
Delete Line 56-57 and changed Men and women show differences in mental and emotional tendencies. In general, men show logical, progressive, and aggressive tendencies, while women have strong maternal instincts and emotional and passive tendencies [5]. Therefore, it is necessary to study gender separately, and since most caregivers are women, we would like to study female care workers. |
|
5. lines 66-67- are the caregivers nurses? If yes, then decide which term to use: caregivers or nurses? There is a difference between the terms caregivers and nurses. Kindly clarify. |
Clarified words. use as a caring instead of a nurse Subjects of my research are only caregivers. Thank you. |
|
6. I suggest that you re write your introduction by building a strong background on the gap regarding the psychological well-being of the caregivers followed by descriptions and discussions of the factors you have identified such as emotional labor, empathy ability, wisdom, and the socio-demographic profiles. It is recommended that you cite theoretical background/foundation of these concepts. |
Introduction introduced the study of Maziriri's psychological well-being [4]. Many revisions have been made so that the authors' arguments can be made in line with the research method. In the discussion, we proceeded according to the results. There was no theoretical framework exactly consistent with this study, so we referred to a similar paper. Thank you very much for your careful review. |
|
7. Procedures: Lines 136-139 - Kindly explain further what do you mean by the statement. It is not clear. |
Revised The explanation is as follows. The purpose of the study is to identify factors that affect psychological well-being by using the general characteristics, emotional labor, emotional empathy, and wisdom of home-visiting female caregivers. Using the questionnaire, if a female caregiver agrees to the study, sign a written consent form and fill out a self-reported questionnaire for about 15 minutes. |
|
8. Instruments: Were the questions in English or translated to the vernacular? If yes, kindly describe the procedure of translation and how was it validated and tested for internal consistency? |
The questionnaire is not a question, it's a statement. It is a tool that is already used with high validity in Korean. It is a highly valid and reliable tool that has been studied for caregivers, care worker, or similar age groups. Nevertheless, we checked and used the tool's validity and reliability. |
|
9. Regression analysis was not mentioned under statistical analysis. |
Added The analysis of psychological well-being effecting factors was analyzed using hierarchical regression analysis using the participants' general characteristics, emotional labor, empathy ability, and wisdom. |
|
10. In table mean- the labels did not identify which is the psychological well-being. Is it the mean scores? If yes, what are the interpretation of the mean scores? Do they have low/poor, moderate, or high/good psychological well-being? |
Added Table 1 labels psychological well-being and adds the meaning of the mean value. |
|
11. Table 1- I suggest that you include a dollar equivalent of the won currency. What do you mean by yes or no under religion? Also what do you mean by education received? |
Changed In Table 1, 200 million won= 1520 dollars, so added dollars(won) Religion-> Have, None No have Education-> Number of education on psychological well-being in one year |
|
12. Lines 228-229 - Kindly review the statement. |
Changed ~~more than 1520 dollars per month had higher levels of psychological well-being than those who earned less than 1520 dollars monthly. |
|
13. Table 2- I suggest the you organize the scales by categorizing them according to your main variables to have a clear picture. Also to add an interpretation ( e.g. low, moderate, high) of the scores for each of the categories (emotional labor, empathy, wisdom, psychological well-being). |
added the degree of the variable to Table 2 and descripted it. In other words, the participants' emotional labor was low, the empathy was moderate, the degree of wisdom was above average, and the degree of wisdom was high in the mutual relationship. Psychological well-being was moderate, and psychological well-being in autonomy was low.
|
|
14. Table 3- what are the relationships? |
Added explanation In other words, the less emotional labor the participants has, the higher the cognitive empathy ability, emotional empathy ability, and total empathy ability, and the higher the degree of wisdom, the higher the participants’ psychological well-being. |
|
15. Discussion- lines 295-308 - I do not find it relevant to the focus of your study. Your discussion should revolve around support on your hypothesis (correlations and predictive models to psychological well-being) and not deviate from the variables you have identified. |
Thank you so much I deleted line 295-308, and changed Kim et al. [6] had a psychological well-being level of 3.33 points in a study of home-visiting caregivers during COVID-19, similar to the results of this study. Since the score of psychological well-being of caregivers was average, improvement methods as a psychological program for caregivers caring for elderly people with dementia [23] are required to continuously increase the score. |
|
16. Figure 1- The model does not reflect the hypothesis and finding of the study. |
Changed figure 1
|
|
17. The conclusion sounds like a summary of the results. Therefore, what can you conclude from these findings? |
Revised conclusion These results will recognize the relationship between general characteristics, emotional labor, empathy and wisdom in relation to the job and social and psychological health of home-visiting caregivers in the future, and serve as basic data for emotional labor, empathy and wisdom development programs. It provided opportunities for repetitive research through the expansion of subjects. In addition, in the practice of caring for the elderly, the head of the center can use it as data for preparing systems and policies that can increase the psychological well-being of female caregivers.
|
|
General comments: The organization and grammar of the article should be considered. There are repetitive concepts, and grammar lapses that make many of the sentences confusing.There should be consistency and clear link among the variables being studied, strong support of the study hypothesis, and consistency in following through from the introduction to the discussion. Thank you and good luck. |
As you pointed out, I revised the introduction significantly. The results also include meaning questions, and the discussion part has been modified to convey meaning. Changed all areas in red collar |

Reviewer 2 Report
The study is well executed, designed and reported. The topic is highly relevant. The research problem is properly formulated, and the aims are stated ckearly. The research design is applicable and the statistical methods are explained well. The conclusion is clear, and the recommendations reflect the aim of the study.
The following minor corrections should be made:
I do have the following comments/suggestions/concerns. 1) Moderate English editing are required. I include 3 examples. 1.1) Under point 5 Conclusions in line 6 the following should be corrected: ----"variables was It was 71%" to ---"variables was 71%" 1.2) Under point 1 Introduction line 4: the following should be corrected: ----" major the Organization" to ----" major , Organization". Delete the "the" 1.3) Under point 1 Introduction second paragraph line 1: the following should be corrected: ---"A part of the people's such desire" to --- "A part of the peoples desire." Delete "such" 2) 21 of the 39 References are older than 5 years. I recommend that the authors update these references where possible.Author Response
|
Reviewer 2 |
|
|
The study is well executed, designed and reported. The topic is highly relevant. The research problem is properly formulated, and the aims are stated ckearly. The research design is applicable and the statistical methods are explained well. The conclusion is clear, and the recommendations reflect the aim of the study. |
Thank you very much for your careful review to make it a good paper. As you pointed out, I revised it hard.
Thank you so much. |
|
The following minor corrections should be made: I do have the following comments/suggestions/concerns.
1) Moderate English editing are required. I include 3 examples. |
I asked the English editing company to review it again. Thank you. |
|
1.1) Under point 5 Conclusions in line 6 the following should be corrected: ----"variables was It was 71%" to ---"variables was 71%" |
Deleted ‘was it’ |
|
1.2) Under point 1 Introduction line 4: the following should be corrected: ----" major the Organization" to ----" major , Organization". Delete the "the" . |
Deleted ‘the’ |
|
1.3) Under point 1 Introduction second paragraph line 1: the following should be corrected: ---"A part of the people's such desire" to --- "A part of the peoples desire." Delete "such" |
Deleted ‘such’ |
|
2) 21 of the 39 References are older than 5 years. I recommend that the authors update these references where possible. |
As you pointed out, I wanted to revise them (7,10,13,18,19,27,38,41,47) to the latest one as much as possible. Unfortunately, some of the old references (28,15,29,30,31,32,5,6) could not be modified because they are related to tools. I will definitely consider it in future studies. Thank you. |

Reviewer 3 Report
Thank you for the opportunity to review this manuscript.
The paper is very interesting, current, and more studies are needed to help us improve and understand the level of psychological well-being of caregivers, because of the burden of care that this entails.
I would like to know what kind of training the caregivers have, if this is regulated and necessary to work, since one of the criteria was to have more than 6 months of experience.
It is not clear to me why you have excluded male caregivers, why it was a criterion and if you could explain it more.
I think it would be interesting to point out some of the limitations of the study and proposals for the future as a separate section.
Congratulations for your work.
Author Response
|
Reviewer 3 |
Thank you very much for your careful review to make it a good paper. As you pointed out, I revised it hard. |
|
The paper is very interesting, current, and more studies are needed to help us improve and understand the level of psychological well-being of caregivers, because of the burden of care that this entails. |
Thank you so much. I think so, too. This is because I think that the well-being of nursing care workers who provide a lot of care will ultimately improve the health and quality of life of patients and those in demand. |
|
I would like to know what kind of training the caregivers have, if this is regulated and necessary to work, since one of the criteria was to have more than 6 months of experience. |
Many professional occupations have an apprenticeship period of about three months or more after first joining the company. Since the caregiver studied at the academy for about 240 hours, it is judged that it takes about six months to heal the elderly or vulnerable people and work as a professional. This system is a unique system in Korea and can only serve as a true caregiver if it reaches this level, so it is based on six months of experience to prevent confusion in the results of the study <inserted> The curriculum of caregivers is a total of 240 hours, with 80 hours of theory related to nursing, 80 hours of practice, and 80 hours of practice. After obtaining the certificate, she must have about six months of experience recognized in practice to facilitate job performance and become a true caregiver, so in order to reduce confusion in the result of study, the criteria were set as subjects with more than six months of practical experience. . |
|
It is not clear to me why you have excluded male caregivers, why it was a criterion and if you could explain it more. |
added On the other hand, there are very few male caregivers currently working in the field, and their duties are different from those of female caregivers. Mainly, male caregivers belong to long-term care hospitals rather than performing home-visiting care, and perform bathing and movement of male elderly people. Therefore, this study only targets female care workers who visit home and take care of them, and if they have at least six months of working experience, they have a high understanding of the job of care workers and can perform practical tasks properly |
|
I think it would be interesting to point out some of the limitations of the study and proposals for the future as a separate section. |
I didn't put this title separately, but I added it at the end of the discussion as reviewed by the reviewer. Thank you.
The limitations and contributions of this study are as follows. Since convenience was sampled in the selection of subjects for this study and the age group is limited to the middle and old age, it is necessary to expand the region and participants, and expand the age group. In addition, male caregivers have been excluded, so it is necessary to study male caregivers. Using the results of this study, it revealed the factors of education and income among general characteristics as major factors in the psychological well-being of female caregivers, and brought about theoretical expansion in that it included emotional labor, empathy, and wisdom. In addition, the results can be used for research for the psychological well-being of female caregivers. In addition, in practice, for the psychological well-being of female caregivers, it can be used as basic data for the development and application of intervention programs using emotional labor, empathy, and wisdom.
|

Reviewer 4 Report
Thank you for the opportunity to review this manuscript. The comments are as follows:
Abstract: the study design was not mentioned in the abstract.
Introduction:
L 27 – 29: You need a citation to support the percentages provided.
Materials and Methods:
Mention the study design!
L120 – L 121: These are not exclusion criteria since all of them were included in the inclusion criteria.
Overall, the study was well conducted and presented.
Author Response
|
Reviewer 4 |
|
|
Abstract: the study design was not mentioned in the abstract.
|
Thank you very much for your careful review to make it a good paper. As you pointed out, I revised it hard.
Added in abstract Research design is a descriptive correlational study. |
|
Introduction: L 27 – 29: You need a citation to support the percentages provided.
|
Added reference As of 2022, the elderly population aged 65 or older accounted for 17.5% of the Korean population, and it is expected that this will continue to increase and reach 20.6% in 2025, making Korea a super-aged society [1]. |
|
Materials and Methods: Mention the study design!
|
Added 2.1. Research design Research design is a descriptive correlational study that analyzes the effects of general characteristics, emotional labor, empathy, and wisdom of female caregivers on psychological well-being.
|
|
L120 – L 121: These are not exclusion criteria since all of them were included in the inclusion criteria. |
Exclusion criteria, L120 – L 121: deleted. |

Round 2
Reviewer 1 Report
Dear Authors,
Thank you for submitting the revised manuscript.
I have seen most of the review comments addressed; however, there are still some issues:
1. The title says "effects" but the results in the abstract and the body of the manuscript says "Influencing factors", and in other parts "effecting factors", etc. Decide whether the appropriate word is 'effect', 'influencing factors', 'association'. Whichever should match the statistics analysis done and be consistently used throughout the manuscript.
2. The word 'therefore' has been used extensively and also inappropriately to some extent. Substitute the word with another.
3. Lines 49-52 seem not relevant to the context of caregivers.
4. Lines 54-66 - Rewrite to avoid repetition of the same idea. The ideas presented are redundant.
5. Line 132- Is the intervention be only for the caregivers with more than 6 months experience? If not, I suggest to remove the phrase... with more than six months experience in the sentence.
6. Table 3- add a note on what is the basis for the level of significance
7. Discussion - The discussion was a restatement of the results and followed by specific recommendations. Very minimal literature was used to support and discuss the thesis of the study.
8. Conclusion- Presentation was similar to the discussion part.
In general, there is a need to consult an English editor to improve not only grammar but more on the organization and presentation of ideas.
Thank you.
Author Response
|
Reviewer 1 |
Thank you very much for carefully reviewing my paper and making numerous modifications to improve its quality. |
|
1. The title says "effects" but the results in the abstract and the body of the manuscript says "Influencing factors", and in other parts "effecting factors", etc. Decide whether the appropriate word is 'effect', 'influencing factors', 'association'. Whichever should match the statistics analysis done and be consistently used throughout the manuscript. |
In order to match statistical analysis and maintain consistency in terms, we modified it to effect and affecting. |
|
2. The word 'therefore' has been used extensively and also inappropriately to some extent. Substitute the word with another. |
Thank you for pointing that out. I deleted it or modified it to accordingly, consequently, and thus. |
|
3. Lines 49-52 seem not relevant to the context of caregivers. |
Deleted that sentences (line 49-52) |
|
4. Lines 54-66 - Rewrite to avoid repetition of the same idea. The ideas presented are redundant. |
Deleted line 57-69 |
|
5. Line 132- Is the intervention be only for the caregivers with more than 6 months experience? If not, I suggest to remove the phrase... with more than six months experience in the sentence. |
Removed “with more than six months of practical experience” |
|
6. Table 3- add a note on what is the basis for the level of significance |
Added under table 3 Level of significance: p < 0.05 |
|
7. Discussion - The discussion was a restatement of the results and followed by specific recommendations. Very minimal literature was used to support and discuss the thesis of the study. |
The method of describing the discussion is to present the results of this study and write it using the supporting research accordingly, so I described it like that. Added references. Please treat me kindly The discussion part has also been revised a lot. |
|
8. Conclusion- Presentation was similar to the discussion part.In general, there is a need to consult an English editor to improve not only grammar but more on the organization and presentation of ideas. |
Revised the conclusion As a result of identifying the factors affecting the psychological well-being of home-visiting caregivers, it was found that among the general characteristics, continuous education, monthly income, emotional labor, empathy ability, and wisdom were significant factors. Accordingly, in order to increase the psychological well-being of home-visiting female caregivers, it is necessary to develop and implement an intervention program to reduce emotional labor and improve empathy ability and wisdom. Additionally, if measures are taken to improve the continuous education and salary system of caregivers, it will help to increase the psychological well-being of female caregivers. This study provides fundamental data for establishing measures to improve the psychological well-being of caregivers in the future by identifying the factors that affect the psychological well-being of home-visiting female caregivers. Moreover, it can be utilized as fundamental data for the preparing systems and policies that can enhance the psychological well-being of female caregivers by the head of the center in the practice of caring for the elderly.
As you pointed out, we reviewed totally the grammar and composition with the editor |
Round 3
Reviewer 1 Report
Dear authors,
Thank you for your hardwork in revising the manuscript. The paper has been improved, however, you missed to write your hypothesis. You can include it, right after the aim in the introduction part and the decision in the conclusion part.
Thank you.
Author Response
|
Reviewer 1 |
Thank you very much for your careful review and consideration. |
|
you missed to write your hypothesis. You can include it, right after the aim in the introduction part and the decision in the conclusion part. |
Inserted it after aim in introduction, and in front of conclusion. The hypothesis of this study was that the general characteristics, emotional labor, empathy ability, and wisdom of female caregivers will affect psychological well-being. |
